# OsPP2C09 Is a Bifunctional Regulator in Both ABA-Dependent and Independent Abiotic Stress Signaling Pathways

**DOI:** 10.3390/ijms22010393

**Published:** 2021-01-01

**Authors:** Myung Ki Min, Rigyeong Kim, Woo-Jong Hong, Ki-Hong Jung, Jong-Yeol Lee, Beom-Gi Kim

**Affiliations:** 1Division of Metabolic Engineering, National Institute of Agricultural Sciences, RDA, Jeonju-si 54874, Korea; mkmin66@gmail.com (M.K.M.); rigyeong02@gmail.com (R.K.); jy0820@korea.kr (J.-Y.L.); 2Graduate School of Biotechnology & Crop Biotech Institute, Kyung Hee University, Yongin 17104, Korea; hwj0602@khu.ac.kr (W.-J.H.); khjung2010@khu.ac.kr (K.-H.J.)

**Keywords:** PP2CAS bifunction, dreb regulation, aba-dependent/independent pathway

## Abstract

Clade A Type 2C protein phosphatases (PP2CAs) negatively regulate abscisic acid (ABA) signaling and have diverse functions in plant development and in response to various stresses. In this study, we showed that overexpression of the rice ABA receptor OsPYL/RCAR3 reduces the growth retardation observed in plants exposed to osmotic stress. By contrast, overexpression of the OsPYL/RCAR3-interacting protein OsPP2C09 rendered plant growth more sensitive to osmotic stress. We tested whether OsPP2CAs activate an ABA-independent signaling cascade by transfecting rice protoplasts with luciferase reporters containing the drought-responsive element (DRE) or ABA-responsive element (ABRE). We observed that OsPP2CAs activated gene expression via the cis-acting drought-responsive element. In agreement with this observation, transcriptome analysis of plants overexpressing OsPP2C09 indicated that OsPP2C09 induces the expression of genes whose promoters contain DREs. Further analysis showed that OsPP2C09 interacts with DRE-binding (DREB) transcription factors and activates reporters containing DRE. We conclude that, through activating DRE-containing promoters, OsPP2C09 positively regulates the drought response regulon and activates an ABA-independent signaling pathway.

## 1. Introduction

As plants inevitably face adverse environmental conditions such as drought, high salt, and extreme temperatures, they have evolved complex signaling networks that respond to environmental cues and balance their resources between promoting growth and mounting tolerance to inauspicious conditions. Abscisic acid (ABA) plays a central role during abiotic stress responses, in addition to influencing various aspects of plant development and growth, such as seed dormancy, leaf abscission, growth inhibition and fruit ripening [1,2,3].

ABA signal transduction starts with the formation of a complex between the ABA receptor PYRABACTIN RESISTANCE1/PYR-like/REGULATORY COMPONENT OF ABA RECEPTOR (PYR/PYL/RCAR) and Clade A Type 2C protein phosphatases (PP2CAs), resulting in the inactivation of PP2CAs and the activation of SNF1-related protein Kinase 2 (SnRK2). Activated SnRK2s in turn activate ABA-responsive element (ABRE)-binding factors (ABFs). ABA-induced gene expression is mediated through the binding of ABFs to the ABRE in target promoter regions and subsequent transcriptional activation [4,5,6].

Yamaguchi-Shinozaki et al., (1992) cloned nine genes responsive to desiccation stress in Arabidopsis (*Arabidopsis thaliana*). *RD29A* and *RD29B* are two closely related genes that respond to drought and high salt conditions. The *RD29A* promoter region contains multiple cis-acting elements such as the dehydration-responsive element/C-repeat (DRE/CRT) and ABRE. *RD29A* expression can therefore respond to both ABA-dependent and -independent signaling pathways through the ABRE or DRE/CRT cis-element, respectively. By contrast, the *RD29B* promoter region lacks a DRE/CRT sequence and is activated only by the ABA-dependent signaling cascade [7]. Dehydration-responsive element binding factor (DREB) transcription factors have been determined to regulate the gene expression in ABA-independent signaling pathways such as cold and drought [8,9].

ABA-dependent signaling is mainly regulated by phosphorylation and dephosphorylation of the signaling components such as SnRK2s and ABFs. However, several studies have shown that DREB degradation is a key regulatory step during ABA-independent signaling. For example, DREB2A, which plays an important role in the ABA-independent pathway in response to drought stress, is marked for degradation by the 26S proteasome via the E3 ubiquitin ligase DREB2A-INTERACTING PROTEIN1 (DRIP1) [10]. In addition, *DREB2A* expression is regulated by GROWTH-REGULATING FACTOR7 (GRF7), which binds to the *DREB2A* promoter and represses transcription [6,11]. The expression of another DREB factor, Arabidopsis *DREB1A*, is influenced by the Myc transcription factor, INDUCER OF CBF EXPRESSION1 (ICE1), which is sumoylated by the SUMO E3 ligase SIZ1, and marked for degradation by the E3 ubiquitin ligase, high expression of osmotically responsive gene1 (HOS1) [8,12,13]. How phosphorylation and dephosphorylation of DREB contributed to its activation was unclear [9] until the recent report that showed phosphorylation at the negative regulatory domain (NRD) destabilizes DREB2A [14].

In this study, we elucidate a new biological role for OsPP2CAs in the ABA-independent pathway and show that they act as a bifunctional regulator of the crosstalk between the ABA-independent and dependent pathways.

## 2. Results

### 2.1. Transgenic Rice Overexpressing OsPYL/RCAR3 Are Insensitive to Osmotic Stress

In previous studies, plants overexpressing the cytosolic ABA receptors OsPYL/RCARs showed increased tolerance to abiotic stress and hypersensitivity to ABA and osmotic stress [3,15,16,17,18]. We generated transgenic rice lines overexpressing a monomeric ABA receptor, OsPYL/RCAR3, and named the line A30 (Appendix A). As expected for an ABA receptor, the A30 transgenic line exhibited increased tolerance to abiotic stress and hypersensitivity to ABA in terms of young seedling growth compared to the control (empty vector control, named A8). In addition, A30 plants were smaller than A8 control plants even in the absence of exogenous ABA application (Appendix A). A30 seedlings did not show hypersensitivity to an increase in NaCl concentrations (Figure 1C) as measured by seedling growth. However, A30 plants demonstrate hyposensitive phenotypes when mannitol was applied in concentrations over 200 mM (Figure 1A–D).

To characterize the effects of OsPYL/RCAR3 overexpression, we introduced OsPYL/RCAR3-HA into rice protoplasts by transient transfection together with a reporter construct bearing the rice Rab16A promoter driving the expression of firefly luciferase (pRab16A::fLUC), which is normally induced by ABA and osmotic stress. We then treated protoplasts with ABA or polyethylene glycol (PEG) to mimic osmotic stress conditions (Figure 1E). Normalized firefly luciferase signal clearly increased in protoplasts overexpressing OsPYL/RCAR3 (4.9-fold increase over mock transfection). Treating OsPYL/RCAR3-overexpressing protoplasts with ABA had no effect on the luciferase signal, as it rose 4.7-fold over the mock sample. However, treating OsPYL/RCAR3-overexpressing protoplasts with with 25% PEG resulted in a 53% reduction in luciferase activity relative to mock-treated overexpressing protoplasts (Figure 1E). These results thus raise the possibility that OsPYL/RCAR3 may inhibit ABA-independent osmotic stress signaling.

### 2.2. Stress-Inducible OsPP2CAs Interact with OsPYL/RCAR3 to Activate Abiotic Stress Responses

OsPP2CAs activate ABA signaling through their ABA-mediated interaction with ABA receptors. OsPYL/RCAR3 belongs to a group of monomeric ABA receptors (Appendix A). As a first step toward identifying OsPP2CAs that are candidates that interact with OsPYL/RCAR3 under osmotic stress conditions, we profiled *OsPP2CA* expression levels in response to ABA treatment or high concentrations (200 mM) of mannitol. *OsPP2C06*, *OsPP2C09* and *OsPP2C30* showed the highest induction out of all other *OsPP2CA* genes by mannitol and ABA treatment (Figure 2A,B).

We then determined the subcellular localization of these three OsPP2CAs by fusing their coding sequences to the *green fluorescent protein* (GFP) gene. OsPP2C06 localized to the cytosol, whereas OsPP2C09 and OsPP2C30 localized to the nucleus (Appendix A). The interaction between OsPYL/RCAR3 and OsPP2C06, OsPP2C09, or OsPP2C30 was confirmed by yeast two-hybrid assays, even in the absence of ABA in the growth medium (Appendix A). Bimolecular fluorescence complementation (BiFC) and co-immuno-precipitation assays further corroborated the interaction between the ABA receptor OsPYL/RCAR3 and the three OsPP2C proteins (Appendix A).

To examine the effects of *OsPP2CA* overexpression, we introduced the abiotic stress reporter (*pRab16A::fLUC*) together with increasing concentrations of plasmid DNA for the effectors *OsPYL/RCAR3, OsPP2C06, OsPP2C09*, or *OsPP2C30* into rice protoplasts by transient transfection (Figure 2C). As expected, fLUC activity, reflecting *Rab16A* promoter activation, increased with the amount of *OsPYL/RCAR3* plasmid DNA used during transfection. Unexpectedly, OsPP2Cs also activated transcription from the *Rab16A* promoter in a dose-dependent manner, although measured fLUC activity levels were much lower than with OsPYL/RCAR3. We thus conclude that the three OsPP2Cs OsPP2C06, OsPP2C09, or OsPP2C30 can act as positive regulators of stress signaling.

### 2.3. OsPP2CAs Induce Gene Expression through the DRE Cis-Element

*Rab16A* is responsive to both ABA and osmotic stress because its promoter contains both DRE and ABRE elements. To establish whether OsPP2CAs activate *Rab16A* expression through the ABA-dependent or ABA-independent signaling pathway, we constructed two new reporter constructs, in which firefly luciferase was driven by promoter fragments that bore only the DRE (*pDRE::fLUC*) or both the DRE and ABRE elements (*pABRE-DRE::fLUC*); both constructs also contained a minimal promoter consisting a TATA box transcription start site (Figure 3A). DNA for both reporters was introduced into rice protoplasts together with empty vector or with effector plasmid DNA for *OsPYL/RCAR3*, *OsPP2C09*, or *OsPP2C30*. We then examined the level of reporter activation for each effector with and without ABA. The *pABRE-DRE::fLUC* clearly responded to ABA, and fLUC activity rose further when OsPYL/RCAR3 was overexpressed (Figure 3B). By contrast, overexpression of *OsPP2C09* or *OsPP2C30* suppressed the induction of luciferase activity resulting from ABA treatment. These results demonstrate that OsPYL/RCAR functions as a positive regulator whereas OsPP2C09 and OsPP2C30 are negative regulators of ABA signaling, as previously reported.

In contrast to the *pABRE-DRE::fLUC* reporter, the *pDRE::fLUC* construct did not respond to ABA. Overexpression of *OsPP2C09* or *OsPP2C30* increased fLUC activity in an ABA-independent manner, but *OsPYL/RCAR3* overexpression did not induce fLUC activity (Figure 3B). To validate this result, we examined fLUC activity derived from the *pDRE::fLUC* reporter as a function of increasing effector DNA amounts during transfection. We used *OsPYL/RCAR3* as a control, with a characteristic weakly positive slope between detectable fLUC activity and transfected *OsPYL/RCAR3* DNA amounts (*S* = 0.098). By contrast, transfecting rice protoplasts with increasing amounts of the *OsPP2C09* effector resulted in a 14-fold rise in fLUC activity over background, with a steep slope (*S* = 1.70). The gradual overexpression of either *OsPP2C06* or *OsPP2C30* also induced fLUC activity more than *OsPYL/RCAR3*, with moderately strong slopes (*S* = 0.20 and *S* = 0.39, respectively) (Figure 3C). These results show that overexpression of OsPP2CAs can activate signaling mediated by the *cis*-acting DRE.

In addition, we examined the possible relationship between OsPYL/RCAR3 and OsPP2CAs in the activation of transcription through the *cis*-acting DRE. We measured fLUC activity from the *pDRE::fLUC* reporter when one OsPP2CA and increasing amounts of OsPYL/RCAR3 plasmid effector DNA were co-transfected into rice protoplasts. *OsPYL/RCAR3* overexpression counteracted the increase in fLUC activity normally associated with *OsPP2CA* overexpression and did so in a dose-dependent manner (Figure 3D). We obtained the same result when we used *pOsRab16A::fLUC* albeit the responses were weaker than with *pDRE::fLUC* (Appendix A). Thus, we conclude that overexpression of *OsPYL/RCAR3* inhibited the action of OsPP2Cs through the ABA-independent *cis*-acting DRE.

### 2.4. Transgenic Rice Plants Overexpressing OsPP2C09 Are Hypersensitive to Osmotic Stress

To determine the function of OsPP2CAs under osmotic stress, we examined transgenic rice lines (named C18) overexpressing *OsPP2C09* (Figure 4A–C). Growth of C18 seedlings was more sensitive to osmotic stress than the control A8 line, when properly normalized to the mock treatment. Indeed, in the absence of mannitol, C18 seedlings were 10% taller than A8 seedlings. However, C18 seedlings appeared slightly smaller than A8 seedlings when grown in the presence or 150 mM or 200 mM mannitol (Figure 4A), although this difference was not significant. We therefore converted shoot growth to relative lengths by normalizing each value to the 0 mM control growth values in Figure 4C. The two independent transgenic lines C18-1 and C18-2 seedlings thus exhibited a growth deficit of 12.3% p (C18-1) or 24.2% p (C18-2) in the presence of 150 mM mannitol and 13.6% p (C18-1) or 20.5% p (C18-2) in the presence of 200 mM mannitol when compared to the A8 control line. These results suggest that C18 seedlings are more sensitive than A30 seedlings (overexpressing *OsPYL/RCAR3*) to osmotic stress.

### 2.5. Transcriptomic Analysis of Rice Overexpressing OsPP2CA09 during Osmotic Stress and ABA Treatment

OsPP2CAs can regulate gene expression through activation of *cis*-DREs in promoters but also repress gene expression resulting from ABA signaling. To identify genes regulated by OsPP2C09, we dissected the transcriptome of rice seedlings (C18-2-1) overexpressing OsPP2CA09 and the control rice cultivar Dongjin (DJ) treated with 200 mM mannitol or 10 µM ABA for 24 h using deep sequencing of the transcriptome (RNAseq). Read numbers were normalized to FPKM (fragments per kb of transcripts per million mapped reads) and then compared with those of non-treated control seedlings. We detected 1631 and 1598 genes that were upregulated more than two-fold in response to 200 mM mannitol or 5 μM ABA treatment, respectively (Figure 5A). Of these, 828 genes were upregulated in both conditions, and are given in Appendix A.

Next, we searched for DRE (A/GCCGAC) and ABRE (ACGTGT/GC) *cis*-elements in the 1-kb upstream promoter regions of these upregulated genes using AME (Analysis of Motif Enrichment) on the MEME website (Multiple EM for Motif Elicitation, http://meme-suite.org/index.html) [19]. Among the 1631 genes upregulated by mannitol, 453 genes contained the DRE alone (276 genes) or both the ABRE and the DRE (177 genes) in their promoters. None of the 1631 promoter sequences contained only the ABRE element. By contrast, among the 1598 genes upregulated by ABA, 458 promoter sequences contained *cis*-ABRE element alone (354 genes) or both cis-ABRE element and *cis*-DRE (104 genes). DRE was detected alone in 141 promoters (Figure 5B). The selected genes are listed in Appendix A. These results showed that low osmotic stress imposed by mannitol upregulated a limited set of genes, but the *cis*-ABRE element that activates transcription in response to ABA signaling is lacking in their promoters.

Next, we compared the expression levels of the 276 genes whose promoters only contained the *cis*-DRE in C18-2-1 and DJ control seedlings. If one of the functions of OsPP2C09 is to activate gene expression via the *cis*-DRE, we would expect most of the 276 genes to exhibit higher expression in DJ seedlings treated with 200 mM mannitol relative to C18 seedlings grown under the same conditions. For visualization, we generated a heatmap of expression values (in FPKMs) for these 276 genes. Most genes were more strongly expressed in C18-2-1 than in DJ (Figure 5C).

We next compared FPKM values for all genes and all upregulated genes in C18 and DJ treated with mannitol (Figure 5D). The linear regression slope was 0.78, indicating that most genes were generally more highly expressed in the DJ control seedlings than in C18. However, the regression slope between FPKM values of genes upregulated by mannitol was 1.22, in agreement with our heatmap result. Moreover, focusing on FPKM values of genes whose promoters contained either *cis*-DRE and *cis*-ABRE or *cis*-DRE alone, the corresponding regression slopes were 2.11 and 2.18, respectively. These results showed that the presence of *cis*-DREs was associated with higher expression in C18 seedlings overexpressing OsPP2C09 relative to its wild-type parental line. Thus, OsPP2C09 may participate in the activation of gene expression through *cis*-DREs in response to low osmotic stress.

In addition, we examined the expression of several genes in control DJ, A30, or C18 seedlings by RT-qPCR. As previously mentioned, the *Rab16A* promoter contains both a *cis*-DRE and a *cis*-ABRE, whereas the *LIP9* and *OsDREB1C* promoters bear only *cis*-DREs. *Rab16A* transcript levels were highly induced in DJ and C18 seedlings grown in the presence of 250 mM mannitol. *Rab16A* transcript reached higher levels in C18 seedlings subjected to mannitol stress for 24 h compared to that in DJ. By contrast, *Rab16A* transcript levels were greatly induced in A30 seedlings following treatment with 10 μM ABA, and showed a slight and delayed rise in C18, possibly caused by OsPP2C09 (Figure 6A). LIP9 and OsDREB1C expression levels increased to a greater extent in C18 seedlings exposed to 250 mM mannitol than in the DJ control, whereas their transcript levels decreased in A30 following this treatment, likely due to OsPYL/RCAR3-mediated inhibition of OsPP2C. Finally, *LIP9* and *OsDREB1C* transcripts were weakly induced by 10 μM ABA in C18 (Figure 6B,C). Together, these results corroborated our transcriptome analysis above.

### 2.6. OsPP2CAs Regulate the Transcriptional Activity of OsDREBs

Our combined results thus far have revealed that OsPP2CAs activate transcription via the *cis*-DRE in response to osmotic stress. Thus, we inferred that OsPP2CAs might regulate the activity of DREB transcription factors directly or indirectly. To test this possibility, we examined the interaction potential between OsPP2CAs and OsDREBs. We selected three OsDREBs (OsDREB1A, OsDREB1B, and OsDREB1C), based on their expression in rice leaf blades in microarray datasets (Appendix A). Due to the demonstrated auto-activity of OsDREB and OsPP2Cs, we did not perform a yeast two-hybrid first, but rather conducted a BiFC with OsDREBs and OsPP2Cs. We used OsPYL/RCAR3 as a negative control because it shared the same subcellular localization (nucleus) as OsDREBs, but is known not to interact with these transcription factors (Appendix A). The N-terminus of the fluorescent protein Venus (VN) was fused with OsPP2Cs and OsPYL/RCAR3; the C-terminus of Venus (VC) was fused with OsDREBs. The resulting constructs were then introduced into rice protoplasts in chosen combinations. We also co-transformed a mCherry marker as control for transfection, when no fluorescence signal from Venus was detectable, and normalized Venus fluorescence signal to mCherry fluorescence (Figure 7A,B). All three OsDREBs interacted with OsPP2C09 and OsPP2C30. However, OsPP2C06 failed to complement Venus fluorescence, indicating a lack of interaction between OsDREBs and OsPP2C06. We confirmed the interaction of OsPP2C09 and OsPP2C30 with OsDREBs by a co-immunoprecipitation assay. OsDREB1A showed the strongest interaction, whereas OsDREB1B showed the weakest interaction with OsPPC09 and OsPP2C30 (Figure 7C).

To test whether OsPP2CAs activate OsDREBs, we focused on OsPP2C09 because it showed the strongest interaction with OsDREBs. We tested OsPP2C09-mediated activation of OsDREB1A, OsDREB1B, or OsDREB1C using the *pDRE::fLUC* reporter in transiently-transfected rice protoplasts. First, we established that fLUC activity (as a proxy for activation mediated by the *cis*-DRE) increased when *OsDREBs-HA* effector plasmid DNA was co-introduced into the protoplasts in a dose-dependent manner (Figure 8A). Next, we generated an OsPP2C09 interference line (OsPP2C09-i) in which Gly-139 was replaced by Asn. This point mutant was reported to interfere with ABA signaling as it abrogates phosphatase activity (Sheen et al., 1998). The OsPP2C09-i point mutant also interrupted ABA signaling in our rice protoplast luciferase assay (Figure 8B).

We next monitored the transcriptional activity of *OsDREB1s* co-transfected with *OsPP2C09-i* or *OsPP2C09*. OsDREB1A-mediated activity clearly increased as a function of OsPP2C09 plasmid effector DNA dosage, compared to other OsDREB1s (Figure 8C). However, OsPP2C09i did not show severe effects on luciferase activity (Figure 8C,D). When transfected cells were incubated for up to 20 h, the luciferase activity induced by OsPP2C09 increased sharply, indicating a synergistic response between OsPP2C09 and OsDREB1A. However, OsPP2C09i did not increase DREB activity under the same conditions (Figure 8D). Instead, OsPP2C09 showed a synergistic effect on the activation of *cis*-DRE by OsDREB1s, especially with OsDREB1A. The OsPP2C09i point mutant, lacking phosphatase activity, did not show activation of *cis*-DRE.

## 3. Discussion

ABA plays central roles in tolerance to abiotic stress such as drought, cold, high temperatures, and salinity. Other signaling pathways, such as the DREB-mediated pathway, also participate in plant abiotic stress responses. Thus, abiotic stress responses may be largely classified into two groups: ABA-independent and ABA-dependent signaling cascades [6].

PP2CAs are well-known negative regulators of ABA signaling. However, their expression is induced significantly not only during treatment with exogenous ABA, but also during osmotic stress [20]. This phenomenon puzzled scientists for a long time and was just explained as a feedback regulation between ABA signaling and osmotic stress (Figure 2A,B) [21]. However, in this study we suggest that PP2CAs, induced by osmotic stress, can activate drought-responsive regulons from the ABA-independent signaling branch and act as positive regulators of signaling.

### 3.1. Nucleus-Localized PP2CAs Regulate Diverse Abiotic Stress-Related Transcription Factors

ABA-dependent signal transduction relies heavily on phosphorylation/dephosphorylation mechanisms of the regulatory proteins [22]. Plant genomes harbor several thousands of kinase genes; whose products are responsible for the phosphorylation of very specific substrates [23]. By contrast, plant genomes contain genes encoding only a few hundred phosphatases, which might have broader substrates than kinases. PP2CAs are major negative regulators of the ABA signaling pathway that function by dephosphorylating SnRK2-type kinases [24].

In total, nine rice PP2CAs can be clearly divided into two groups, based on their subcellular localization (nucleus and cytosol) [25]. Cytosol- or membrane-localized OsPP2CAs might regulate the activity of several second messengers and transporters and play diverse functions. Such PP2Cs, as for example Arabidopsis ABA INSENSITIVE1 (ABI1) and ABI2, are the main regulators and initial signal transducers of the ABA signal [26]. However, the six nucleus-localized PP2CAs potentially can modulate transcription factors, directly or indirectly. Bhatnagar et al., 2017 have reported that PP2CAs can interact with OsbZIP and directly regulate this transcription factor independently of SnRKs, suggesting that PP2CAs can directly regulate transcription factors [27]. In addition, Mizoi et al., 2019 reported that phosphorylation of the NRD region makes DREB2A unstable; this result raised the possibility that phosphatases like OsPP2C09 may affect the stability or activity of DREBs [14]. We therefore investigated the abiotic stress responsive transcription factors DREBs during ABA-independent signaling, although their phosphorylation status during such treatment was obscure in this report. We propose here that one of the nucleus-localized PP2CA, OsPP2C09, might regulate DREB1 activity. Thus, OsPP2C09 can modulate ABA-dependent signaling as well as ABA-independent signaling reversibly.

### 3.2. PP2CAs Might Act as Hubs in ABA-Dependent and -Independent Signaling Crosstalk

ABA, drought, osmotic and cold stress signals all sense environmental cues via different receptors [28]. Thereafter, the receptors transduce these signals to downstream signaling components such as kinases, phosphatases and so on [29]. Transcriptional regulation is finally enacted by transcription factors binding to *cis*-regulatory elements in the promoters of target genes [30]. The receptor(s) for osmotic stress is not known yet, but might initiate a faster response than the ABA signaling cascade. Even though the two signaling pathways have largely distinct signaling components, both rely on PP2CAs, which may target several common substrates and regulate their phosphorylation/dephosphorylation status in response to both signals.

In this and previous studies, PP2CAs are induced quickly by abiotic stresses and can then activate DREBs regulons, which include ABA biosynthesis and growth regulators. Thus, PP2CAs might act as one of the early regulators of abiotic stress responses.

After recognition of the initial osmotic stress, ABA content increases in the cell. This rise in ABA levels may amplify the responses necessary for adaptation to abiotic stresses, and PP2Cs may provide feedback regulation to balance stress tolerance responses (which limit growth) and sustained growth (which limits stress responses) [6,21]. Abiotic stress signaling components are shared by both ABA-independent and -dependent signaling pathways.

We hypothesize that OsPP2CAs might have several substrates in both ABA-dependent and ABA-independent signaling, and antagonistically regulate the two signaling pathways. Thus, PP2CAs might function as a central signaling hub in both abiotic stress-signaling pathways.

## 4. Materials and Methods

### 4.1. Plant Materials and Generation of Transgenic Rice

We surface-sterilized rice seed (*Oryza sativa* cv. Dongjin) with 70% ethanol for 30 s and 50% bleach for 40 min, respectively. After 5 washes with distilled water, seeds were sown on half-strength Murashige and Skoog [31] medium pH 5.8 (supplemented with 0.4% phytagel) for 7 days under long-day conditions (16 h light and 8 h darkness) at 28 °C. To generate transgenic rice, we amplified the coding sequence of *OsPYL/RCAR3* (locus number Os02g15640) or *OsPP2C09* (locus number *Os01g62760*) via PCR from first-strand cDNAs with specific primers (5-topo-RCAR3 and 3-RCAR3, or 5-topo-PP2C09 and 3-PP2C09, listed in Appendix A) and AccPrime ^TM^
*Pfx* DNA polymerase using the provided protocol. The PCR conditions were 2 min denaturing and 35 cycles (denaturing 95 °C for 30, annealing at 55 °C for 30 s, and extending at 68 °C for 40 s). The PCR products were inserted into pENTR/D-Topo (Invitrogen, Carlsbad, CA, USA) and then transferred to the pGA2897 vector via Gateway LR recombination. The resulting *pUbi:OsPYL/RCAR3* or *pUbi:OsPP2C09* constructs, in which *OsPYL*/*RCAR3* or *OsPP2C09* was placed under the control of the constitutive maize *UBIQUITIN* promoter, was transformed into Agrobacterium (*Agrobacterium tumefaciens*) strain LBA4404 via electroporation. We generated transgenic rice plants using the Agrobacterium-mediated co-cultivation method and selected the transformants based on hygromycin resistance and before transferring them to the greenhouse (Toki et al., 2006).

### 4.2. Post-Germination Assays and Stress Treatment

We plated surface-sterilized dehulled seeds on half-strength MS medium supplemented with hygromycin (40 mg/L). Then, 3 days later, we transferred the seedlings to half-strength MS medium supplemented with 5 µM ABA, 200 mM NaCl, 200 mM Mannitol in square Petri dishes (125 × 125 × 20 mm). Seedling growth was measured 10 days after transfer. This experiment was repeated three times independently.

### 4.3. RT-PCR and Quantitative PCR

For RT-qPCR analysis, we synthesized first-strand cDNAs from 5 μg total RNA using SuperScript III reverse transcriptase (Invitrogen, Carlsbad, CA, USA) using the provided protocol. A 1:40 dilution of the cDNAs was used for RT-qPCR. The amplification parameters were as follows: 15 min of denaturation and enzyme activation at 95 °C; followed by 40 cycles of 95 °C for 5 s, 60 °C for 15 s, and 72 °C for 30 s; with a final step performed at 65–95 °C (1 °C/s) for melting curve analysis. The amplified signals were detected using a MyiQ real-time PCR system (Bio-Rad Laboratories, Hercules, CA, USA) using SYBR Premix Ex Taq^TM^ (TOPrealTM qPCR 2X PreMix, www.enzynomics.com, Yuseong-gu, Daejeon, Korea). The data were normalized based on the expression of rice *UBIQUITIN5*, and the relative gene expression was analyzed using the 2^−^^ΔΔ^^Ct^ method or the 2^−^^Δ^^Ct^ method. Primer sequences used for RT-qPCR analysis are listed in Appendix A.

### 4.4. Luciferase Assay Using Rice Protoplasts

We isolated rice protoplasts using the protocol described by Kim et al., (2015). For the luciferase assay, firefly luciferase (fLUC) and renilla luciferase (rLUC) plasmids and effector plasmids were introduced into purified rice protoplasts as previously described [32]. We introduced all plasmid DNAs (5 μg of *pRab16a:fLUC, pABRE-DRE:fLUC,* or *pDRE:fLUC* with 0.5 μg of *pUbi:rLUC* (transformation control) and indicated effector plasmids) into rice protoplasts by the PEG transfection method. After 15–20 h incubation, fLUC and rLUC activity was detected using a dual-luciferase assay kit (Promega, Madison, WI, USA) and a GloMax 96 Microplate Luminometer (Promega, USA), according to the manufacturer’s instructions.

### 4.5. Subcellular Localization and BiFC

To observe interactions between proteins using bimolecular fluorescence complementation (BiFC), we introduced gene fragments into pENTR-D-TOPO vectors (Invitrogen, Carlsbad, CA, USA) and then transferred them to their destination vectors by LR recombination (Promega, Madison, WI, USA) [25]. pGEM-gw-VC vector was used for fusion of Venus Carboxyl-terminus (VC) with OsDREB1A, OsDREB1B, and OsDREB1C, pGEM-gw-VN vector used for fusion of Venus Amino-terminus (VN) with OsPP2C06, OsPP2C09, osPP2c30, and OsPYL/RCAR3. Generated plasmids were introduced into rice protoplasts as indicated pairs using the PEG-mediated method [32]. The ER-mCherry reporter was used as internal control. Fluorescence signals were captured using a Leica TCS SP8 laser scanning confocal microscope (Leica Microsystems, Wetzlar, Germany). The combination of excitation wavelength/detection range of emission for Venus signals was 488 nm (solid state laser)/ with a bandpass of 505–561. Signal intensities on captured images were analyzed using Leica Application SuiteX provided by the manufacturer (Leica Microsystems, Wetzlar, Germany) with default settings (threshold 30%, background 20%).

### 4.6. Co-Immunoprecipitation

To perform co-immunoprecipitation experiments, we used GFP-trap (Chromotek, Planegg, Germany) precipitate GFP-fusion proteins. All used genes were PCR-amplified with specific primers (Appendix A) and inserted into pENTR/D-topo vectors (Invitrogen, Carlsbad, CA, USA). We then recombined these entry clones with the vectors pGEM-gw-GFP for GFP fusions and pGEM-gw-3xHA for HA tagging [25], using LR recombination (Invitrogen, Carlsbad, CA, USA). The indicated constructs were introduced into rice protoplasts using the PEG-mediated method, and the transformed protoplasts were incubated at 28 °C for 20 h. Cellular extracts from transformed protoplasts extracted in immunoprecipitation buffer (150 mM NaCl, 50 mM Tris-HCl at pH 7.5, 1 mM EDTA, 2 mM EGTA, 2 mM MgCl_2_, 0.5% NP40, 0.5% Triton X-100, and 1x protease inhibitor cocktail (complete ULTRA tablet, Roche, Indianapolis, IN, USA)) were incubated with pre-cleaned GFP-trap beads at 4 °C for 2 h. After 5 washes with immunoprecipitation buffer, the precipitated proteins, together with GFP-trap, were subjected to SDS-PAGE with mid-range protein marker (Elpis Biotech, Daejeon, Korea) and immunoblot analysis. Precipitated GFP and HA-tagged proteins were detected with anti-GFP rabbit antibody (Life Technologies, Carlsbad, CA, USA) and anti-hemagglutinin (HA) rat antibody (Roche, Indianapolis, IN, USA), respectively.

### 4.7. Transcriptome Analysis

For the transcriptomic analysis, we prepared the total RNA from 14 day-old plants grown on 1/2 MS medium. C18 (OsPP2C09-overexpressing plant) and its control Dongjin plants were prepared with or without ABA or mannitol treatment for 24 h. Total RNA was extracted from shoots and purified using the RNeasy Mini Kit (Qiagen, Hilden, Germany). Quality control was conducted with the Agilent Technologies 2100 Bioanalyzer (Agilent Technologies, Santa Clara, CA, USA). The libraries for sequencing were prepared using a TruSeqRNA Sample Prep Kit v2 (Illumina, San Diego, CA, USA), following the manufacturer’s instructions. The sequencing of the libraries was performed using a HiSeq 4000 system (Illumina, San Diego, CA, USA) generating single-end 101-bp reads. The trimmed reads were mapped to IRGSP (v. 1.0) and assembled into transcripts. The read counts were determined using the StringTig program, and then normalized with the DESeq2 program [33]. The promoter sequences of upregulated genes were obtained from The Rice Annotation Project Database annotated data on OsNipponbare-Reference-IRGSP-1.0. The motifs searches were performed in http://meme-suite.org/tools/ame with indicated conditions [19]. The heatmap image was constructed with MeV program [34]. The graphs were constructed using GraphPad Prism6 program (GraphPad Software, San Diego, CA, USA).

## Figures and Tables

**Figure 1 ijms-22-00393-f001:**
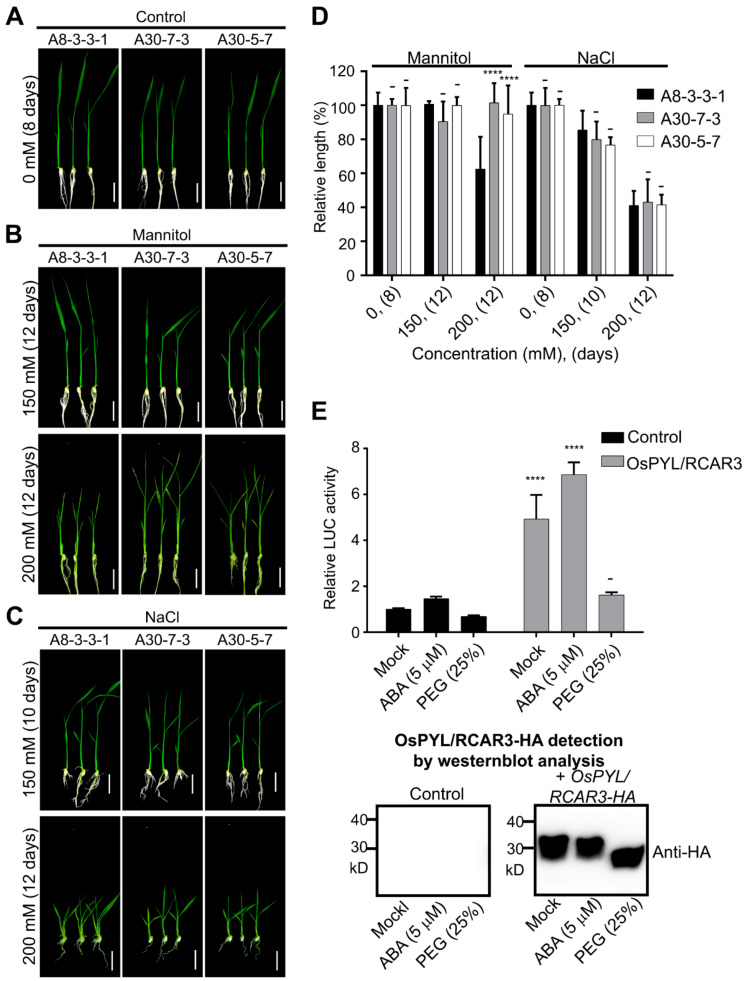
A30 plants, overexpressing OsPYL/RCAR3, have reduced sensitivity to mannitol-imposed osmotic stress. (**A**–**C**), Seedling growth test. A8 and A30 seedlings were grown in growth medium alone (**A**) or containing mannitol (**B**) or NaCl (**C**). Scale bar: 2 cm. (**D**) Measurement of relative shoot length compared to each mock (0 mM). X axis indicates mannitol or NaCl concentrations, and growth period, (in d), in brackets. Two-way ANOVA was performed with A8 plants as controls. ****: *p* < 0.0001, -: not significant. (**E**) Effect of OsPYL/RCAR3 overexpression on luciferase activity from the *OsRab16a* promoter in rice protoplasts. Control: *pRab16a: LUC* alone. Data presented as mean ± standard deviation (SD), *n* = 3. Two-way ANOVA was performed, comparing with control. ****: *p* < 0.0001, -: not significant. OsPYL/RCAR3-HA expression was confirmed by immunoblot analysis with anti-hemagglutinin (HA) rat antibody.

**Figure 2 ijms-22-00393-f002:**
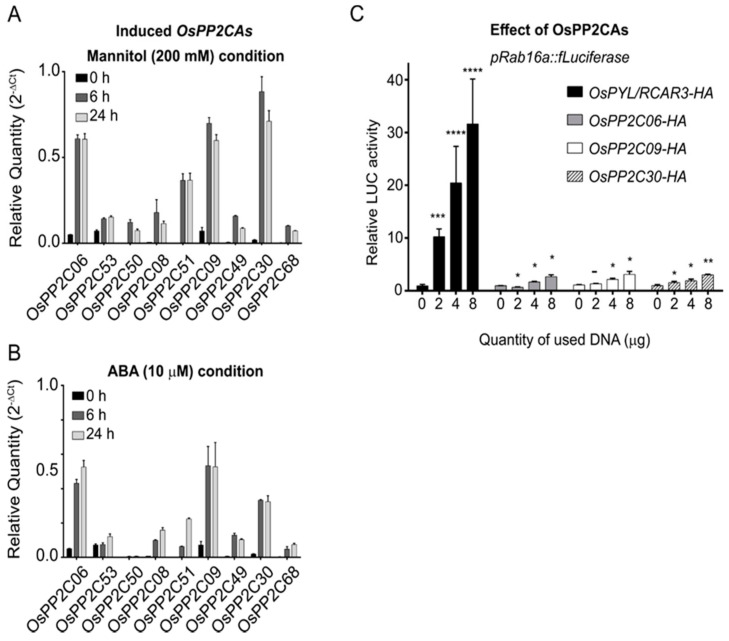
Highly expressed *PP2CAs* in the A30 line activate transcription from the *Rab16A* promoter. (**A**,**B**), relative transcript levels of 9 *PP2CA* genes after exposure of A30 seedlings to 200 mM Mannitol or 10 μM ABA for 0, 6, or 24 h. All values were determined by RT-qPCR and normalized to *UBIQUTIN5* levels. Error bars are means ± SD (*n* = 3). Similar results were obtained in three iterations. (**C**), Effects associated with overexpression of selected *OsPP2C* genes on *Rab16* promoter activity (*pRab16A:fLUC* and *pUbi:rLUC*) introduced into rice protoplasts by PEG transfection. After a 15 h incubation, luciferase activities were detected. All values were normalized to luciferase activity with 0 μg of *OsPP2C* overexpression vector. Error bars are means ± SD (*n* = 3). One-way ANOVAs were performed with comparing to each 0 μg (****: *p* < 0.0001, ***: *p* < 0.001, **: *p* < 0.01, *: *p* < 0.05, -: not significant).

**Figure 3 ijms-22-00393-f003:**
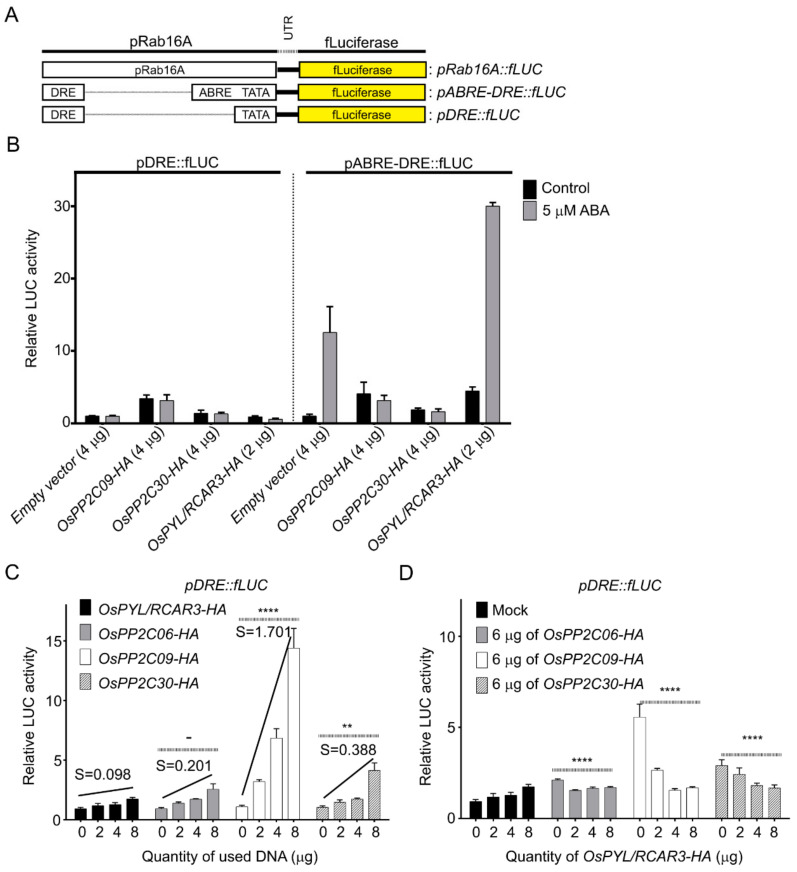
OsPP2Cs activate *Rab16a* transcription via *cis*-DRE. (**A**), Schematics of generated reporter plasmids, *pABRE-DRE::fLUC* and *pDRE::fLUC*. (**B**), Activation test of *pABRE-DRE::fLUC* or *pDRE::fLUC* with effector only or ABA in rice protoplasts. (**C**), Activation of the *pDRE::fLUC* reporter with increasing amounts of the indicated effector DNA during transfection. (**D**), Effects of OsPYL/RCAR3 overexpression on OsPP2C-mediated induction of luciferase activity from the *pDRE::fLUC* reporter. The indicated effector and marker DNAs were introduced into rice protoplasts by PEG-mediated transfection and then incubated for 15 h. Induced luciferase activity was detected with a dual-luciferase reporter assay system. All values are means ± SD (*n* = 3). S is the slope of the trend line. Two-way ANOVAs (main column effect) were performed, comparing to OsPYL/RCAR3-HA values (****: *p* < 0.0001, **: *p* < 0.01, -: not significant).

**Figure 4 ijms-22-00393-f004:**
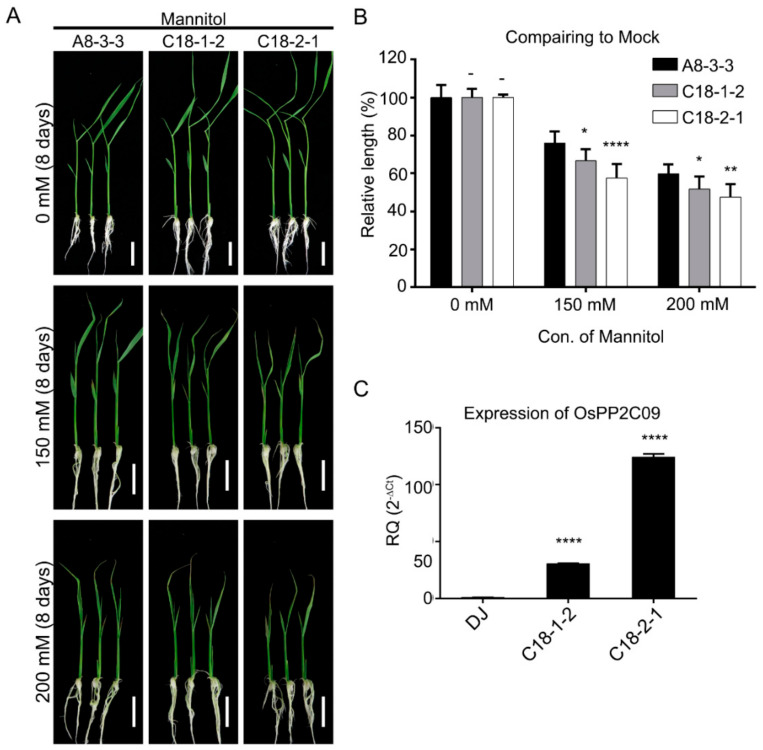
C18 plants, overexpressing *OsPP2C09*, are hypersensitive to osmotic stress. (**A**), A8 and C18 plants transferred after the two-leaf stage to half-strength MS medium containing the indicated mannitol concentrations for 8 d. (**B**), Shoot lengths were measured, with six or more seedlings per replicate. Relative shoot length for A8 and C18 seedlings, normalized to mock control (0 mM mannitol). The values are means ± SD (*n* = 3). The experiment was repeated three times with similar results. Two-way ANOVA was performed comparing with A8 plants (****: *p* < 0.0001, **: *p* < 0.01, *: *p* < 0.05, -: not significant). (**C**), Relative transcript levels of *OsPP2C09* in C18 seedlings. All values were determined by RT-qPCR and normalized to *UBIQUTIN5* before normalization to relative transcript levels in Dong-Jin (DJ), shown as mean of relative quantity and SD. One-way ANOVA was performed (****: *p* < 0.0001).

**Figure 5 ijms-22-00393-f005:**
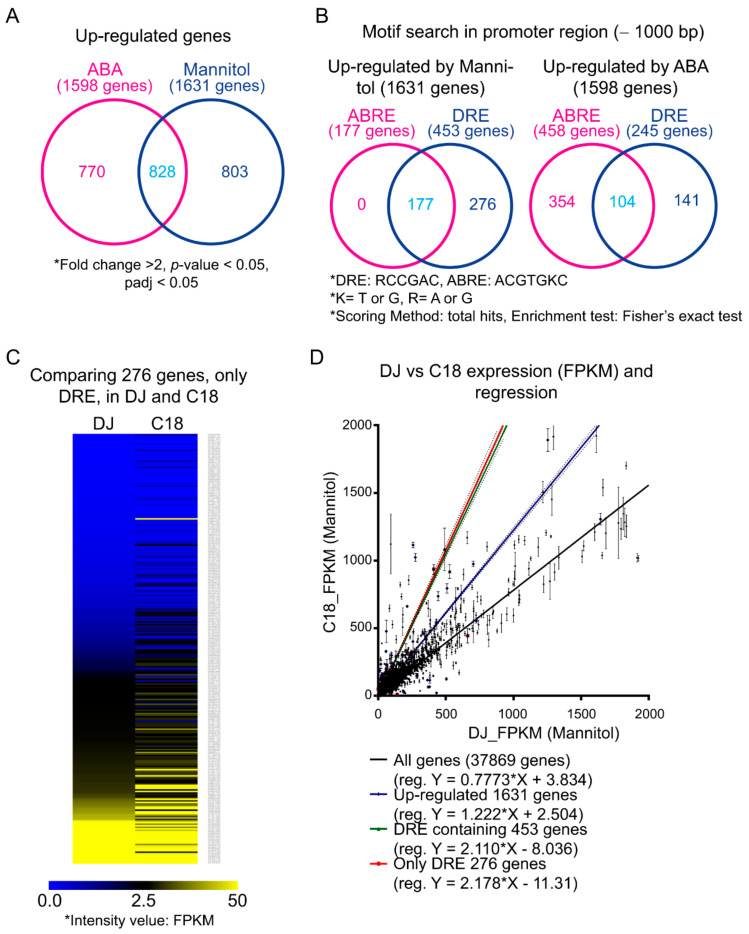
Transcriptome and promoter motif analysis in response to mannitol or ABA treatment. (**A**), Venn diagram of upregulated genes after treatment with 200 mM mannitol or 10 μM ABA. The genes were selected with the indicated criteria (fold-change > 2, *p*-value and padj < 0.05). (**B**), Venn diagrams of genes whose promoters contain DRE and/or ABRE *cis*-acting elements. We performed an AME analysis on the MEME website. Venn diagrams were generated with InteractiVenn (http://interactivenn.net). (**C**), Heatmap of FPKM values for 276 genes containing *cis*-DRE. (**D**), Comparison of FPKMs for the indicated genes between wild-type (DJ) and C18 seedlings treated with 200 mM mannitol. Dotted line: slope of 1. The values are FPKM ± SD (*n* = 3).

**Figure 6 ijms-22-00393-f006:**
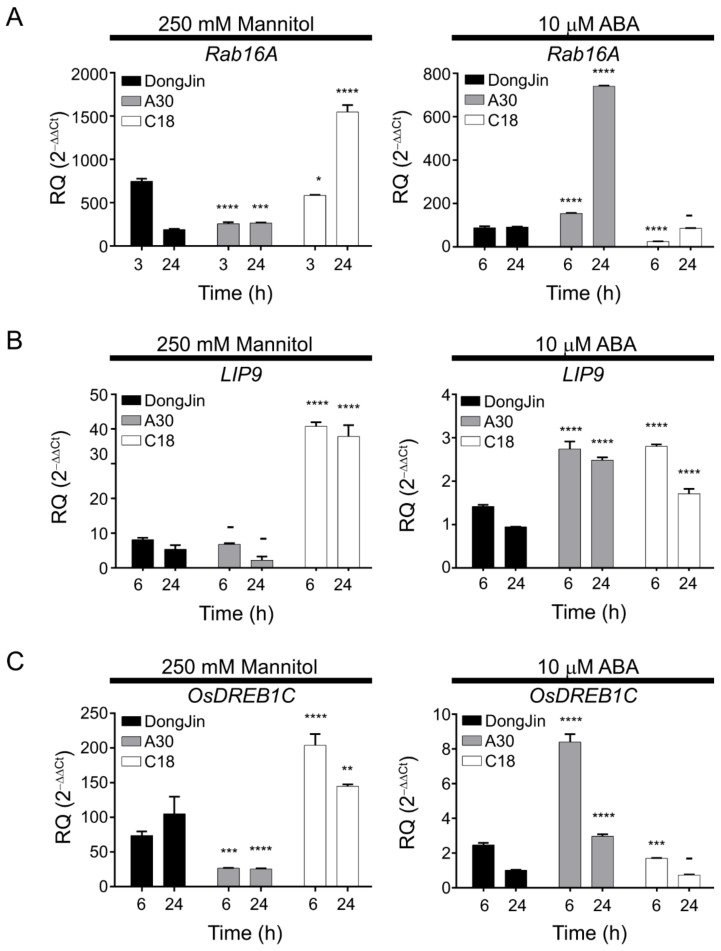
RT-qPCR analysis of three marker genes *Rab16A*, *LIP9*, and *OsDREB1C* in DJ, A30 or C18 seedlings after exposure to 250 mM mannitol or 10 μM ABA. Relative transcript levels of *Rab16A* (**A**), *LIP9* (**B**) or *OsDREB1C* (**C**), determined by RT-qPCR and normalized to *UBIQUITIN5* as internal control. Two-way ANOVAs were performed (****: *p* < 0.0001, ***: *p* < 0.001, **: *p* < 0.01, *: *p* < 0.05, -: not significant). Values are means ± SD (*n* = 3). Similar results were obtained in three replicate experiments.

**Figure 7 ijms-22-00393-f007:**
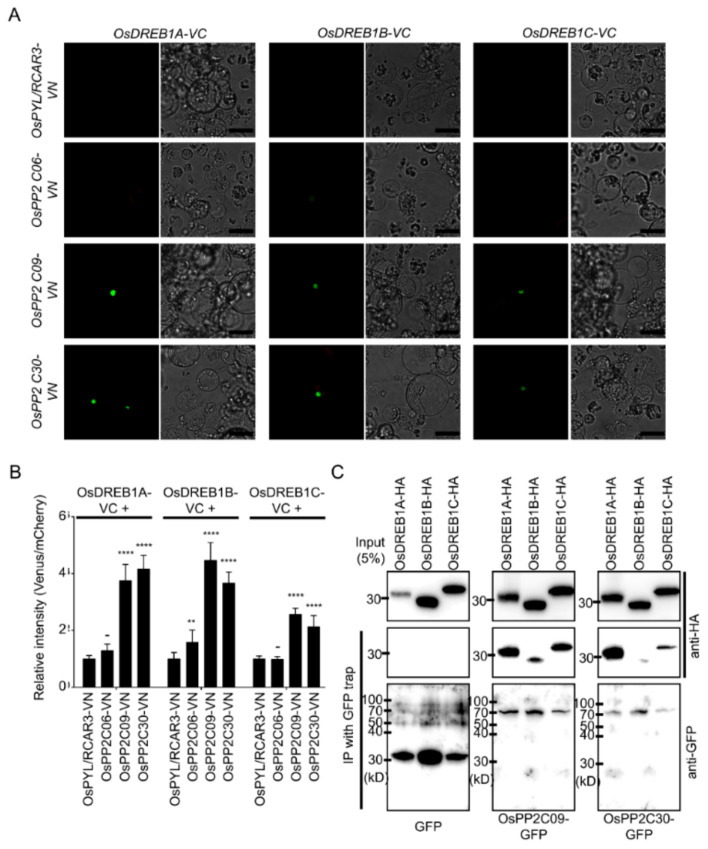
OsPP2CAs interact with OsDREBs. (**A**), Venus *C*-terminus (VC)-tagged OsDREB1A, OsDREB1B and OsDREB1C were examined with Venus *N*-terminus (VN)-tagged OsPYL/RCAR3, OsPP2C06, OsPP2C09, and OsPP2C30 by BiFC analysis. Scale bars: 10 µm. (**B**), BiFC signal intensities normalized to internal control (ER-mCherry). Values are means ±SD (*n* > 20 cells) and normalized to BiFC signal for OsPYL/RCAR3-VN and each OsDREBs-VC. One-way ANOVAs were performed with comparing to signals interacted with OsPYL/RCAR3-VN (****: *p* < 0.0001, **: *p* < 0.01, -: not significant). (**C**), Co-immunoprecipitation analysis. Indicated constructs were introduced into rice protoplasts and GFP-tagged proteins were pull-downed with GFP-trap beads. An immunoblot analysis was performed with anti-GFP rabbit antibodies or anti-HA rat antibodies.

**Figure 8 ijms-22-00393-f008:**
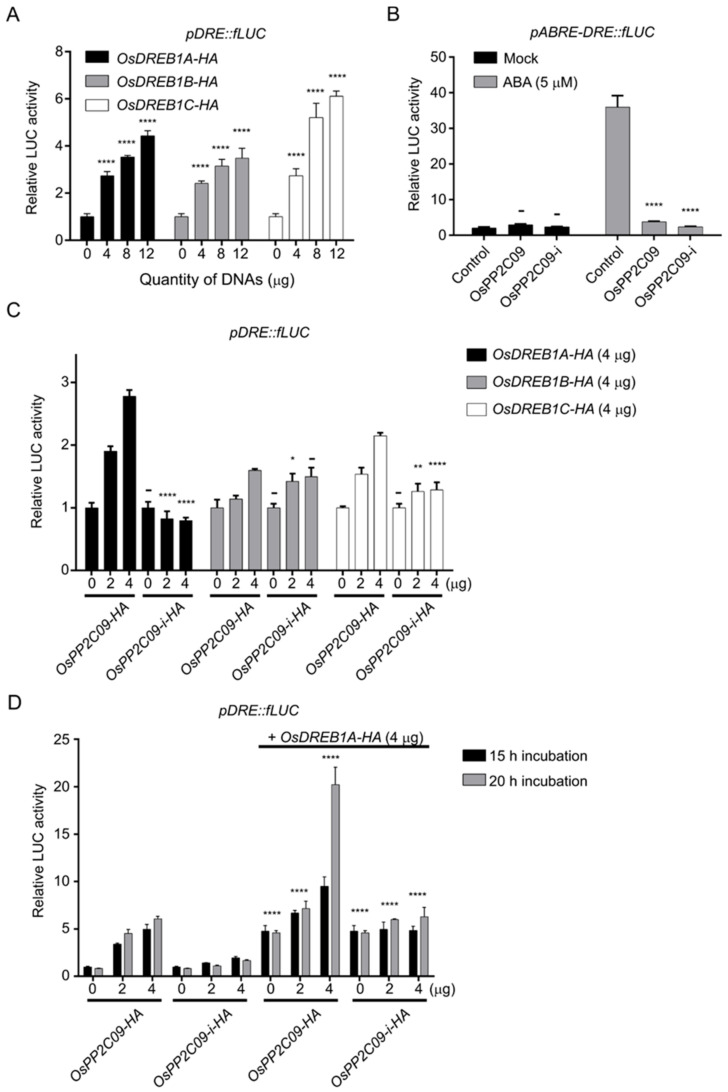
Activation of *pDRE::fLUC* by overexpression of *OsDREB1s.* (**A**), Activation of *pDRE::fLUC* depends on *OsDREB1s* effector plasmid DNA amounts. One-way ANOVAs were performed with comparing to 0 μg of effector DNAs. (**B**), Confirmation of the lack of activation of the *pABRE-DRE::fLUC* reporter by OsPP2C09-i. One-way ANOVAs were performed, comparing to the control. (**C**), Effects of OsDREB1-mediated activation by OsPP2C09 or OsPP2C09-i. Two-way ANOVAs were performed with comparing to WT (OsPP2C09). (**D**), Effects of OsPP2C09 on the activity of OsDREB1A at two time points. Two-way ANOVA was performed, comparing to non-OsDREB1A. The indicated effector and marker DNAs were introduced into rice protoplasts by the PEG-mediated transfection method and then incubated for 15 h or 20 h. Induced LUC activity was detected with a dual-luciferase reporter assay system. All values of panels are means ± SD (*n* = 3). ****: *p* < 0.0001, **: *p* < 0.01, *: *p* < 0.05, -: not significant.

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
