# Peer review of "OsPP2C09 Is a Bifunctional Regulator in Both ABA-Dependent and Independent Abiotic Stress Signaling Pathways"

_ijms, 2021, doi:10.3390/ijms22010393_

Round 1

Reviewer 1 Report

PYL/RCAR and PP2C play central roles of ABA signal transduction. The authors produced overexpressing rice of OsPYL/RCAR3 and this plant showed tolerant to osmotic stress, on the other hand, the reporter assay showed that activation of transcription of Rab16a by PYL/RCAR3 was suppressed under osmotic stress condition compared to non-stress condition or ABA treatment. Then, the authors verified the function of OsPP2C, the interacting protein of PYL/RCAR. The authors demonstrated that OsPP2C09 interacted with OsDREBs and activated transcription of DRE containing genes under osmotic stress, suggesting that OsPP2C09 works both in ABA-dependent and independent pathways.

  1. In “introduction”, the authors mentioned that “How phosphorylation and dephosphorylation of DREB contributes to its activation remains unclear.” Then, in discussion, the authors did not discuss the relationship between PP2C09 and phosphorylation status of DREBs. There was reported that inhibition of phosphorylation stabilizes and activates AtDREB2A (Mizoi et al. JBC 2019). They should be further discussed in DISCUSSION.

  1. Why PEG treated OsPYL/RCAR3-HA signal was smaller than those of mock and ABA treated in fig 1E?

  1. In figure 1E, the reporter assay showed that activation of transcription of Rab16a by PYL/RCAR3 was suppressed under osmotic stress condition compared to non-stress condition or ABA treatment. And, the authors revealed antagonistic relationship between OsPYL/RCAR3 and OsPP2CA in the activation of transcription through DRE genes (Fig3D). Does the OsPP2CAs suppress the activation of transcription of full-length pRab16A by PYL/RCAR3 in a dose-dependent manner?

4. In the lane 74, the authors mentioned plants overexpressing

Author Response

Reviewer 1

Comments and Suggestions for Authors

PYL/RCAR and PP2C play central roles of ABA signal transduction. The authors produced overexpressing rice of OsPYL/RCAR3 and this plant showed tolerant to osmotic stress, on the other hand, the reporter assay showed that activation of transcription of Rab16a by PYL/RCAR3 was suppressed under osmotic stress condition compared to non-stress condition or ABA treatment. Then, the authors verified the function of OsPP2C, the interacting protein of PYL/RCAR. The authors demonstrated that OsPP2C09 interacted with OsDREBs and activated transcription of DRE containing genes under osmotic stress, suggesting that OsPP2C09 works both in ABA-dependent and independent pathways.

  1. In “introduction”, the authors mentioned that “How phosphorylation and dephosphorylation of DREB contributes to its activation remains unclear.” Then, in discussion, the authors did not discuss the relationship between PP2C09 and phosphorylation status of DREBs. There was reported that inhibition of phosphorylation stabilizes and activates AtDREB2A (Mizoi et al. JBC 2019). They should be further discussed in DISCUSSION.

- Thank your commentary for unconfirmed references (Mizoi et al). We added this reference to the introduction and discussion part.

  1. Why PEG treated OsPYL/RCAR3-HA signal was smaller than those of mock and ABA treated in fig 1E?

- I think that the remaining PEG affected the SDS-PAGE separation of the protein. This result has been checked several times.

  1. In figure 1E, the reporter assay showed that activation of transcription of Rab16a by PYL/RCAR3 was suppressed under osmotic stress condition compared to non-stress condition or ABA treatment. And, the authors revealed antagonistic relationship between OsPYL/RCAR3 and OsPP2CA in the activation of transcription through DRE genes (Fig3D). Does the OsPP2CAs suppress the activation of transcription of full-length pRab16A by PYL/RCAR3 in a dose-dependent manner?

- As you suggested, I performed luciferase assay using pOsRab16a::fLUC and attached the results to sup fig 6. Even though the effects of OsPP209 were significantly lower than that of pDRE::fLUC, OsPP2C09 might activate pOsRab16a::fLUC.

  1. In the lane 74, the authors mentioned plants overexpressing

- I think there are some mistakes in the saving the file. This comment was an incomplete sentence.

Reviewer 2 Report

The authors present an interesting study on OsPP2CAs regulates abiotic stress signaling pathway in ABA-dependent and independent manners. 

Here are my comments:

L 395. Could the authors specify the temperature for the germination of the seedlings?

L 397. The PCR conditions are missing.

L 413. The protocol for RNA extraction is missing.

L 421. The authors mentioned that "at least three biological repetitions were performed". How many repetitions where performed for each condition?

Figure 1D. Please correct the y-axis: "Relative length (%)"

Figure 1E: The ladder is missing in the immunoblot and the complete picture should be present in the supplemental figure. The way the figure is display does not provide enough information regarding the specificity of the antibody.

Section 2.5. The authors look at the cis-element in the promoter regions of the genes from 2 different cultivars. Could the authors justify the specificity of cis-elements analysis since they used the promoter sequences from the reference genome (which is different from the cultivars used in the experiment).

Figure 7C: same comment as Figure 1E. the ladder is missing.

Author Response

Reviewer 2

Comments and Suggestions for Authors

The authors present an interesting study on OsPP2CAs regulates abiotic stress signaling pathway in ABA-dependent and independent manners.

Here are my comments:

L 395. Could the authors specify the temperature for the germination of the seedlings?

- A general incubation temperature of 28 degrees was used. Temperature is presented in manuscript.

L 397. The PCR conditions are missing..

- I wrote PCR conditions.

L 413. The protocol for RNA extraction is missing.

- The method provided by the manufacturer was used. It was mentioned in manuscript.

L 421. The authors mentioned that "at least three biological repetitions were performed". How many repetitions where performed for each condition?

- Those were indicated in the legend.

Figure 1D. Please correct the y-axis: "Relative length (%)"

- It was corrected.

Figure 1E: The ladder is missing in the immunoblot and the complete picture should be present in the supplemental figure. The way the figure is display does not provide enough information regarding the specificity of the antibody.

- It was corrected.

Section 2.5. The authors look at the cis-element in the promoter regions of the genes from 2 different cultivars. Could the authors justify the specificity of cis-elements analysis since they used the promoter sequences from the reference genome (which is different from the cultivars used in the experiment).

-That was a good point. The number of SNPs between cultivar.Nipponbare and Dongjin is 0.45 SNP/kb, which we think the genotype is close enough. We don't think there will be a big impact on statistical analysis. A reference that analyzed SNP between Nipponbare and Dongjin was attached.

(Jeong, I.-S., et al. (2013). "SNP-based analysis of genetic diversity in anther-derived rice by whole genome sequencing." Rice 6(1): 6.)

Figure 7C: same comment as Figure 1E. the ladder is missing.

- It was corrected

Round 2

Reviewer 1 Report

Dear Authors,

Thank you for the corrections, the manuscript looks much better.

Author Response

Thank you for your consideration!

Reviewer 2 Report

The authors answered almost all my comments. However, the original ladder/marker is missing on the figures 1E and 7C, also the reference of the ladder/marker used was not provided in the methods. The authors provided the original pictures but in a non publishable format and the readers will not have access to these information.

Author Response

Thank you for your consideration.

In summary, this revision,

  1. I attached protein maker to western blot figure 1E and figure 7C according to reviewer 2's comment and provided protein marker information in line471.
  2. English proofreading of this paper was performed. Corrected grammar or symbols, and there was no change in meaning or content.

Thank you.